# Facile Synthesis of Antimicrobial Aloe Vera-“Smart” Triiodide-PVP Biomaterials

**DOI:** 10.3390/biomimetics5030045

**Published:** 2020-09-17

**Authors:** Zehra Edis, Samir Haj Bloukh

**Affiliations:** 1Department of Pharmaceutical Sciences, College of Pharmacy and Health Science, Ajman University, Ajman P.O. Box 346, UAE; 2Department of Clinical Sciences, College of Pharmacy and Health Science, Ajman University, Ajman PO Box 346, UAE; s.bloukh@ajman.ac.ae

**Keywords:** *Aloe vera*, antibiotic resistance, biomaterials, antimicrobial activity, biodegradable, surgical site infections, nosocomial infections, polyiodides

## Abstract

Antibiotic resistance is an eminent threat for the survival of mankind. Nosocomial infections caused by multidrug resistant microorganisms are a reason for morbidity and mortality worldwide. Plant-based antimicrobial agents are based on synergistic mechanisms which prevent resistance and have been used for centuries against ailments. We suggest the use of cost-effective, eco-friendly *Aloe Vera Barbadensis* Miller (AV)-iodine biomaterials as a new generation of antimicrobial agents. In a facile, one-pot synthesis, we encapsulated fresh AV gel with polyvinylpyrrolidone (PVP) as a stabilizing agent and incorporated iodine moieties in the form of iodine (I_2_) and sodium iodide (NaI) into the polymer matrix. Ultraviolet-visible spectroscopy (UV-Vis), Fourier transform infrared spectroscopy (FT-IR), x-ray diffraction (XRD), microstructural analysis by scanning electron microscopy (SEM) and energy dispersive spectroscopy (EDS) verified the composition of AV-PVP-I_2_, AV-PVP-I_2_-NaI. AV, AV-PVP, AV-PVP-I_2_, AV-PVP-I_2_-NaI, and AV-PVP-NaI were tested in-vitro by disc diffusion assay and dip-coated on polyglycolic acid (PGA) sutures against ten microbial reference strains. All the tested pathogens were more susceptible towards AV-PVP-I_2_ due to the inclusion of “smart” triiodides with halogen bonding in vitro and on dip-coated sutures. The biocomplexes AV-PVP-I_2_, AV-PVP-I_2_-NaI showed remarkable antimicrobial properties. “Smart” biohybrids with triiodide inclusions have excellent antifungal and promising antimicrobial activities, with potential use against surgical site infections (SSI) and as disinfecting agents.

## 1. Introduction

Antibiotic resistance is a danger for the existence of mankind. Infections caused by multi-drug-resistant microorganisms lead to increasing mortality, morbidity, long-term treatment and economic burden in the United States, European Union and worldwide [1,2]. The World Health Organization (WHO) considers antibiotic resistance as a major threat for human existence [3]. Resistance has emerged through the uncontrolled use of available antibiotics worldwide and missing efforts to replace them with new generations of drugs [4,5].

Delays in the healing process rates, prolonged treatment durations, rising morbidity, mortality and economic burden on the health care system are markers of the urgent need for new antimicrobial agents. The ESKAPE pathogens (*Enterococcus faecium*, *Staphylococcus aureus*, *Klebsiella pneumoniae*, *Acinetobacter baumannii*, *Pseudomonas aeruginosa*, *Enterobacter spp*., and *Escherichia coli*) belong to the most resistant species [6]. Methicillin-resistant *Staphylococcus aureus* (MRSA) is responsible for causing the highest infection-associated deaths in the United States of America [7]. Nosocomial infections are part of the problem in the treatment of severely ill, immunocompromised patients with comorbidities [8,9,10]. The contamination of health care and public settings with microorganisms cause secondary infections and severe complications can be tackled by the use of antimicrobial agents [10,11]. Antimicrobial agents are needed to reduce the impact of nosocomial infections which hamper the recovery process [10,11,12]. Nanotechnology is offering an increasing number of applications for drug-delivery systems, disinfectant coatings, antiseptics and biomaterials with silver as the most popular nanoparticle [13,14,15,16]. New, low-cost, non-toxic antibacterial, antiviral agents need to be developed through facile, environmentally friendly synthesis, which can be affordable even in developing countries [10,11,12,13]. The next pandemic striking mankind, may result in higher mortality rates, if there are no new generation antimicrobial agents ready to support the treatment of patients and to prevent nososcomial infections due to multidrug-resistant pathogens. Nosocomial infections in hospital settings raise concern in the treatment of patients, especially in intensive care units [10,17]. Antimicrobial agents, disinfectants, sanitizers and coatings for surfaces in hospitals, health-care and public settings, as well personal protective equipment (PPE) are needed and life-saving [10,11,12,13,14]. The quest for a new generation of antimicrobial agents should culminate in effective coating agents for masks, gloves, personal protective equipment, surfaces, wound-care products, surgical materials and medical equipment to prevent microbial adsorption [10,11,12,13,14]. There is an urgent need for keeping surfaces, medical devices and equipment in hospitals free of pathogenic microorganisms and biofilm formation, in order to support the healing process and fast recovery of immunocompromised patients [18,19]. Copper and povidone iodine in combination with other agents mitigate the microbial load on surfaces and other antimicrobial applications [14,19,20].

Polyiodides contain [I_2k+n_]^n−^ units with stabilizing donor-acceptor interactions of I_2_ and iodide ions [21,22]. Iodine is an important microbicide and has many applications in the medical field, as commercial disinfectants, and against biohazards [20,23,24,25,26]. Complexed compounds with iodine are interesting materials and can be designed in the form of different polyiodides [27,28,29,30,31,32,33]. Halogen bonding within the polyiodide units are “smart” reservoirs of molecular iodine [22,27,28,29,30,31,32,33]. These reservoirs can be utilized against pathogenic microorganisms through the controlled release of iodine from the complexed halogen-bonded polyiodides [22,27,28,29,30,31,32,33]. Disinfectants like povidone iodine are very effective, but have several drawbacks, like skin irritation, discoloration, pruritis and burning pain on the application side [34]. These side effects are related to the uncontrolled release of molecular iodine I_2_ from the compounds, and can be controlled by designing complexed polyiodides with halogen bonding within suitable complexing agents or polymer matrices [22,27,28,29,30,31,32,33,35]. Linear, symmetrical “smart” triiodides are energetically stable due to their halogen bonding [22,27,28,29]. Upon interaction of these complexed polyiodides with microbial membranes, the structure of the complex is deformed [22,27,28,29]. The deformation causes a change in the halogen bonding and leads subsequently to iodine release on the spot against the microbial membrane [22,27,28,29]. Consequently, less iodine content in the formulation is needed to inactivate pathogens. Decrease of iodine content will also reduce the adverse effects during wound healing [20,22,25]. Microbial inactivation and inhibition of biofilm formation can be achieved effectively with minimized iodine content [20,25]. The polymer polyglycolic acid (PGA), is an absorbable, multifilamented surgical suture utilized for closing wounds [36,37]. PGA is known for having good wound healing properties because it is not irritating and non-toxic [36,37]. PGA is absorbed within 60 to 90 days by surrounding wound tissues [36,37]. Multidrug resistant pathogens cause surgical site infections (SSI) around the wound [36,37]. PGA has a braided structure with surface for microbial attachment and biofilm formation, especially by *S. aureus* and *C. albicans* [36]. Coating PGA sutures with antimicrobial agents may prevent complications caused by multidrug resistant pathogens and promises faster healing processes due to on-site, targeted drug-delivery [36].

Polyiodides in plant-biosynthesized nanocomposites may have the potential for developing new generation antibiotics and antimicrobials. These nanomaterials may disable microbial defense mechanisms, which usually lead to resistance. Recently, we reported on complexes of povidone iodine with trans-cinnamic acid and *cinnamomum zeylanicum* extracts (Cinn) and their use for the biosynthesis of silver nanoparticles (AgNP) [38]. The resulting nanocomposites AgNP-Cinn-PI and AgNP-TCA-PI and their dip-coated polyglycolid acid (PGA) sutures inhibited a selection of microbial reference strains by in vitro agar diffusion methods, confirming their potential to prevent surgical site infections (SSI) [38]. The plant components may increase the inhibition of microorganisms by synergistic effects.

Polyphenols, flavonoids and further plant-based natural products have high inhibition potential against pathogens [39,40,41,42,43,44]. Plant-based antimicrobials depend on synergistic mechanisms among their multiple constituents [40]. This pharmacokinetic synergism enhances their biological activities, but diminishes after purification to a single active compound [40]. The natural combination of all components is needed to ensure the same activity. Nevertheless, plants remain a rich source of promising biosynthesized antimicrobial agents for mankind and are used historically against ailments [39,40,41,42,43,44,45]. *Aloe barbadensis* Miller (AV) is a well-known, low-cost, easy-to-cultivate plant, which even grows in arid regions. The gel inside the leaves of AV consists of more than 98% water and 1–2% bioactive compounds [46]. More than 75 bioactive substances are available in AV, including acemannan, aloin, aloe-emodin, aloesin, aloe-mannan, flavonoids, sterols, amino acids, vitamins, enzymes, and minerals [45,46,47,48]. AV components have promising antimicrobial properties, and many pharmacological applications and are used against many diseases since centuries in many cultures [45,49,50]. Hence, conflicting reports overshadow the biological characterization of AV and its constituents. There is an urgent need for standardization in extraction, storage and antimicrobial testing methods. The biological activities in AV are due to synergism and depend heavily on plant species, age, soil type, its quality and salinity, watering regimen, dehydration, sun exposure, temperature, climate, time and season of harvest [51,52,53]. According to a review in 2019, acemannan is the major bioactive polysaccharide derived from AV gel [49]. Acemannan, AV polysaccharides and anthraquinones revealed antiviral activity against the human influenza A virus (H1N1) [54,55]. Hence, acemannan content depends on species and cultivation conditions, irrigation with seawater, harvest season, growth age and solar irradiation intensities [49]. Acemannan is one of the important antimicrobials within AV, but drying processes heavily reduce the acemannan acetylation content by deacetylation [56]. Aloin is effective in inflammatory processes, cancer and skin diseases [45]. Aloin and aloe-emodin inhibited multi-drug resistant *S. aureus* biofilm formation [57]. Some major antimicrobial constituents like aloin and aloe-emodin can be increased by the control of climatic, geographical conditions and extraction methods [51,58]. These are available in a higher amount in AV in colder climatic regions. AV is sensitive to extreme temperatures, which causes stress on the plant [51,58]. Cold stress leads to higher phytochemical content [51,58]. AV grown in semi-arid to humid subtropical zones have higher aloe emodin contents than tropical and arid climate zones [51,58]. Harvesting in summer renders higher bioactive compounds and antimicrobial properties compared to winter [51,58]. Extraction and drying processes, the temperature used, the solvent, and the amount of bioactive components within the AV gel influence the product quality and inhibition of pathogens [59]. According to a report in 2015, fresh AV gel (10 mL) was only directly dissolved in DMSO (100 mL) and effectively inhibited the multidrug resistant *P. aeruginosa* in burn wound infections [59]. Standardization is needed to avoid the pitfalls and increase the biological activity of the products. This will ensure the proper and effective utilization as antimicrobial agent, especially in prevention and infection control. The AV components are incorporated into polymer matrices for wound dressing and surgical processes, to ensure on-site drug delivery [60]. Recent reports included curcumin into the AV-polymer matrix with improved wound healing properties [60,61]. In one report, curcumin was added to lyophilized AV powder, used on nanofibers consisting of polymers, and delivered topically on the wound tissues [61]. Disc diffusion assays against *S. aureus ATCC 29231*, methicillin resistant *S. aureus ATCC 700699*, *P. aeruginosa ATCC 9027*, and *E. coli ATCC 8739* revealed good inhibition zones [61].

In this work, we encapsulated freshly extracted AV gel with polyvinylpyrrolidone (PVP) and incorporated iodine (I_2_), as well as sodium iodide (NaI) into the polymer matrix. Our biocomposites AV-PVP-I_2_ and AV-PVP-I_2_-NaI and their dip-coated PGA surgical sutures proved to have excellent antifungal properties against *C. albicans WDCM 00054*, and antibacterial activity against five Gram-positive (*S. pneumoniae ATCC 49619*, *S. aureus ATCC 25923*, *E. faecalis ATCC 29212*, *S. pyogenes ATCC 19615*, *Bacillus subtilis WDCM 00003*), and four Gram-negative (*E. coli WDCM 00013*, *P. aeruginosa WDCM 00026*, *P. mirabilis ATCC 29906*, *K. pneumoniae WDCM 00097*) bacterial reference strains. The novelty of our compounds is in the combination of two basic components known through centuries by mankind for their healing properties. Iodine and the natural plant extract of Aloe Vera gel, encapsulated and stabilized by PVP, opens up new dimensions of treatment options in wound healing. The microbicide iodine is stabilized by the polymer biocomplex and released on-site by controlled drug-release, supported by the healing effects of aloe vera. We impregnated the PGA sutures with our biomaterials by simple dip-coating and verified the strong inhibition of *C. albicans* and intermediate action on further pathogens at concentrations of 50 μg/mL and below. The facile, cost-effective and rapid biosynthesis of our biomaterials AV-PVP-I_2_, AV-PVP-I_2_-NaI, along with their antimicrobial action in-vitro and on surgical polyglycolic acid (PGA) sutures, renders them as potential new generation antimicrobial agents. Our hybrids maybe also be suitable as disinfectants, coatings and wound healing agents.

## 2. Materials and Methods

### 2.1. Materials

Polyvinylpyrrolidone (PVP-K-30), Iodine (≥99.0%), sodium iodide (NaI), Sabouraud Dextrose broth, and Mueller Hinton Broth (MHB) were purchased from Sigma Aldrich (St. Louis, MO, USA). Disposable sterilized Petri dishes with Mueller Hinton II agar and McFarland standard sets were obtained from Liofilchem Diagnostici (Roseto degli Abruzzi (TE), Italy). *Aloe vera* leaves (*Aloe barbadensis* Miller, AV) were received from the botanical garden of Ajman University. The bacterial strains *E. coli WDCM 00013* Vitroids, *P. aeruginosa WDCM 00026* Vitroids, *K. pneumoniae WDCM 00097* Vitroids, *C. albicans WDCM 00054* Vitroids, and *Bacillus subtilis WDCM 0003* Vitroids were purchased from Sigma-Aldrich Chemical Co. (St. Louis, MO, USA). *S. pneumoniae ATCC 49619*, *S. aureus ATCC 25923, E. faecalis ATCC 29212, S. pyogenes ATCC 19615 and P. mirabilis ATCC 29906* were obtained from Liofilchem (Roseto degli Abruzzi (TE), Italy). Gentamicin (9125, 30 µg/disc), cefotaxime (9017, 30 µg/disc), chloramphenicol (9128, 10 µg/disc), streptomycin (SD031, 10 µg/disc), and nystatin (9078, 100 IU/disc) were purchased from Liofilchem (Roseto degli Abruzzi (TE), Italy). Methanol (analytical grade) was received from Fisher Scientific (Loughborough, UK). Sterile polyglycolic acid (PGA) surgical sutures (DAMACRYL, 75 cm, USP: 3-0, Metric:2, 19mm, DC3K19) were obtained from General Medical Disposable (GMD), GMD Group A.S., Istanbul, Turkey. Sterile filter paper discs with a diameter of 6 mm were purchased from Himedia (Jaitala Nagpur, Maharashtra, India). Ultrapure water was used. All reagents were of analytical grade and used as obtained.

### 2.2. Preparation of Aloe vera (AV) Extract

Leaves from 3-year-old *Aloe vera* (*Aloe barbadensis* Miller) plants growing under the shade of the trees in the botanical garden of Ajman University campus in Ajman, UAE, were harvested in the morning hours between 8:30–9:00 am, and taken directly into the nearby research laboratory of Ajman University, College of Pharmacy and Health Sciences. The freshly-cut, 38–50 cm long *Aloe vera* (AV) leaves were immediately washed with distilled water to remove sand, then rinsed once with pure methanol, several times with ultrapure water, and dried carefully to remove the remains of the water. The AV leaves were sliced with a knife, and the gel was scrapped out and placed in a blender. This mucilaginous, pure gel was homogenized for 3 min at maximum speed, and transferred into a 1000 mL Erlenmeyer flask. 80 g of the colorless gel was placed into a 250 mL Erlenmeyer, mixed with 80 mL absolute methanol, covered, and stirred at room temperature for 18, 36 and 54 h. After 18, 36 and 54 h batches of 20 mL were taken and centrifuged at 4000 rpm for 30 min (3K 30; Sigma Laborzentrifugen GmbH, Osterode am Harz, Germany). The light-yellow supernatant was kept sealed in darkness at 3 °C for further use.

### 2.3. Preparation of AV-PVP, AV-PVP-I_2_, AV-PVP-I_2_-NaI and AV-PVP-NaI

The stock solution AV-PVP is prepared by dissolving 1g polyvinylpyrrolidone K-30 (PVP) in 10 mL water. Moreover, 2 mL of this solution is added to 4 mL of AV under stirring at room temperature.

AV-PVP-I_2_ is obtained by mixing 4 mL of the stock solution AV-PVP with 0.05 g of iodine previously dissolved in 3 mL methanol under stirring at room temperature.

For the synthesis of AV-PVP-I_2_-NaI, 0.05 g I_2_ and 0.026 g NaI are first dissolved under continuous stirring at room temperature, then added to 4 mL of stock solution AV-PVP.

AV-PVP-NaI is prepared by adding 4 mL of stock solution AV-PVP into a solution of 0.026 g NaI in 3 mL methanol.

### 2.4. Characterization of AV Complexes

The prepared complexes were analyzed by SEM/EDS, X-ray diffraction (XRD), UV-vis, and FT-IR. The analytical methods confirmed the composition of AV-PVP-I_2_ and AV-PVP-I_2_-NaI.

#### 2.4.1. Scanning Electron Microscopy (SEM) and Energy-Dispersive X-ray Spectroscopy (EDX)

The compound AV-PVP-I_2_-NaI was analyzed at 5 kV by scanning electron microscopy (SEM), equipped with energy-dispersive X-ray spectroscopy (EDX), model VEGA3 from Tescan (Brno, Czech Republic). One drop of each compound was dispersed in distilled water, dropped onto a carbon-coated copper grid, and left under ambient conditions until dried. These samples were then coated by a Quorum Technology Mini Sputter Coater with a gold film. The energy dispersive spectroscopy (EDS) analysis system examined the elemental composition of the compounds.

#### 2.4.2. X-ray Diffraction (XRD)

The composition of our compounds AV-PVP-I_2_ and AV-PVP-I_2_-NaI were examined by XRD (BRUKER, D8 Advance, Karlsruhe, Germany). Coupled Two Theta/Theta with time/step of 0.5 s and a step size of 0.03 were measured by Cu radiation (wavelength = 1.54060 A).

#### 2.4.3. UV-Vis Spectrophotometry (UV-Vis)

The absorbance spectrum of AV-PVP-I2, AV-PVP-I2-NaI, and AV-PVP-NaI was recorded by a UV-Vis spectrophotometer model 1800 from Shimadzu (Kyoto, Japan), using the wavelength range from 190 to 800 nm.

#### 2.4.4. Fourier-Transform Infrared Spectroscopy (FT-IR)

The two compounds AV-PVP-I_2_ and AV-PVP-I_2_-NaI were freeze-dried and then studied at 400–4000 cm^−1^ using a FTIR spectrometer (Shimadzu, Koyoto, Japan).

### 2.5. Bacterial Strains and Culturing

Reference microbial strains of *S. pneumoniae ATCC 49619*, *S. aureus ATCC 25923*, *E. faecalis ATCC 29212*, *S. pyogenes ATCC 19615*, *Bacillus subtilis WDCM 0003* Vitroids, *P. mirabilis ATCC 29906*, *E. coli WDCM 00013* Vitroids, *P. aeruginosa WDCM 00026* Vitroids, *K. pneumoniae WDCM 00097* Vitroids, and *C. albicans WDCM 00054* Vitroids were utilized for the antimicrobial testing. These microbial strains were stored at −20 °C and inoculated MHB by adding the respective fresh bacteria and fungi. These suspensions were kept at 4 °C until further use.

### 2.6. Determination of Antimicrobial Properties of AV, AV-PVP-I_2_, AV-PVP-I_2_-NaI, and AV-PVP-NaI

The compounds AV, AV-PVP, AV-PVP-I_2_, AV-PVP-I_2_-NaI, and AV-PVP-NaI were tested against the Gram-positive *S. pneumoniae ATCC 49619*, *S. aureus ATCC 25923*, *S. pyogenes ATCC 19615*, *E. faecalis ATCC 29212*, *B. subtilis WDCM 00003*, *Gram-negative bacteria P. mirabilis ATCC 29906*, *P. aeruginosa WDCM 00026*, *E. coli WDCM 00013*, and *K. pneumoniae WDCM 00097*. The antifungal activities were tested against *C. albicans WDCM 00054*. The batches of 36 and 54 h showed a trend of steadily decreasing antimicrobial activities compared to the 18 h stirring batches and were not specifically mentioned further.

#### 2.6.1. Procedure for Zone of Inhibition Plate Studies

The zone of inhibition plate method was used to test the antimicrobial activities of AV-PVP-I_2_ and AV-PVP-I_2_-NaI against the selected pathogens [62]. The bacterial strains were first suspended in 10 mL MHB and incubated at 37 °C for 2–4 h, while *C. albicans WDCM 00054* was incubated on Sabouraud Dextrose broth at 30 °C. Disposable, sterilized Petri dishes with MHA were evenly seeded with microbial culture of 100 μL adjusted to 0.5 McFarland standard applied with sterile cotton swabs, dried for 10 min and used for the antimicrobial testing.

#### 2.6.2. Disc Diffusion Method

The disk diffusion method according to the Clinical and Laboratory Standards Institute (CLSI) recommendations was employed to test the antimicrobial activities of our samples against the antibiotic discs of gentamycin, chloramphenicol, cefotaxime, streptomycin, and nystatin [63]. Moreover, 2 mL of AV, AV-PVP-I_2_, AV-PVP-I_2_-NaI and AV-PVP-NaI with concentrations of 50 µg/mL, 25 µg/mL, 12.5, and 6 µg/mL were impregnated on sterile filter paper discs and left to dry for 24 h. The fungal strain *C. albicans WDCM 00054* was incubated for 24 h at 30 °C on the agar plates. The diameter of zone of inhibition (ZOI) was measured with a ruler to the nearest millimeter. The diameters of clear inhibition zone around the disc are the markers for the microbial susceptibility towards the tested samples. No inhibition zone proves the resistance of the pathogen. The disc diffusion tests were replicated three times, and the average of these independent experiments are reported.

### 2.7. Preparation of Dip-Coated Sutures and Their Characterization

The three samples AV-PVP-I_2_, AV-PVP-I_2_-NaI and AV-PVP-NaI impregnated on sterile, multifilamented, uncoated PGA sutures were dip-coated separately. The coating was done on 10 suture fragments of approximately 2.5 cm. These sutures were prepared by treating them with acetone and drying at room temperature. They were immersed into each 50 mL of AV-PVP-I_2_, AV-PVP-I_2_-NaI and AV-PVP-NaI solutions (1 mM) 25 °C, and stirred at 130 rpm for 18 h. The color of the sutures changed from blue to brownish-blue. The coated sutures were removed from the solutions, dried for 24 h under ambient conditions and tested in vitro by the zone of inhibition assay against the reference strains *S. pneumoniae ATCC 49619*, *S. aureus ATCC 25923*, *E. faecalis ATCC 29212*, *S. pyogenes ATCC 19615*, *Bacillus subtilis WDCM 00003*, *E. coli WDCM 00013*, *P. aeruginosa WDCM 00026*, *P. mirabilis ATCC 29906*, *K. pneumoniae WDCM 00097*, and *C. albicans WDCM 00054*. The AV-PVP-I_2_-NaI coated sutures and the uncoated PGA were analyzed by SEM and compared.

### 2.8. Statistical Analysis

Data are represented as mean. Statistical significance between groups was calculated by one-way ANOVA. A value of *p* < 0.05 was considered statistically significant. SPSS software (version 17.0, SPSS Inc., Chicago, IL, USA) was utilized for the statistical analysis.

## 3. Results

### 3.1. Characterization of AV Complexes

The samples were analyzed by SEM/EDS, x-ray diffraction (XRD), UV-vis, and FT-IR. These methods confirmed the composition of AV-PVP-I_2_ and AV-PVP-I_2_-NaI. All the complexes are stable and stay homogenous for when stored in the fridge.

#### 3.1.1. Scanning Electron Microscopy (SEM) and Energy-Dispersive X-ray Spectroscopy (EDX)

The sample AV-PVP-I_2_-NaI, the uncoated, braided PGA suture (control) and the dip-coated AV-PVP-I_2_-Na suture were investigated by SEM (Figure 1).

The SEM analysis of the AV-PVP-I_2_-NaI coated PGA suture compared with the uncoated PGA suture demonstrates the even distribution and morphology of AV-PVP-I_2_-NaI deposition (Figure 1b,c). The SEM shows the firm deposition of AV-PVP-I_2_-NaI on the multifilamented, absorbable PGA suture surface without causing changes (Figure 1c).

The complexes AV-PVP-I_2_ and AV-PVP-I_2_-NaI were investigated morphologically by Energy dispersive spectroscopy (EDS) and EDS layered images (Figure 2). The percentage for iodine in AV-PVP-I_2_ and AV-PVP-I_2_-NaI is before and after adding NaI 6.4 and 30.3%, respectively. The EDS layered images confirm the uniform distribution of the components in the two samples (Figure 2).

#### 3.1.2. X-ray Diffraction (XRD)

The XRD analysis of the complexes AV-PVP-I_2_ and AV-PVP-I_2_-NaI demonstrate similar diffraction patterns in Figure 3. The XRD graph in Figure 3a shows two sharp peaks in the range from 2θ = 9° and 14°, indicating that AV-PVP-I_2_ is not completely amorphous. The addition of NaI turns the semicrystalline compound AV-PVP-I_2_ into an amorphous polymer complex, demonstrated by the transformation of the two sharp peaks into broad signals (Figure 3). The XRD reveals a crystalline phase of NaI after adding sodium iodide into AV-PVP-I_2_ in Figure 3b, while the rest of the sample is entirely amorphous. Both graphs show no other phases derived from impurities.

The XRD analysis in Figure 3a shows the presence of crystalline behavior from AV reported earlier [64,65,66]. Rahma et al. [67] studied the XRD of pure PVP and reported broad peaks, indicating amorphous phases, which can be also seen in our XRD analysis (Figure 3). The presence of NaI in AV-PVP-I_2_-NaI disturbs the semicrystalline nature of AV and the amorphous nature of PVP (Figure 3b). Both XRD patterns confirm that the two synthesized biohybrids AV-PVP-I_2_ and AV-PVP-I_2_-NaI are almost amorphous, because only a few crystalline peaks are available in the whole measurement range. Figure 3b confirms the presence of NaI structure in the complexed composite AV-PVP-I_2_-NaI. The XRD analysis results show that the reaction between the AV-PVP polymer, the iodine and sodium iodide rendered a complete biocomposite.

#### 3.1.3. UV-Vis Spectrophotometry (UV-Vis)

The UV-vis spectra of AV, AV-PVP-I_2_ and AV-PVP-I_2_-NaI, and AV-PVP-NaI are shown in Figure 4. The absorption bands of iodine and the triiodide ion in AV-PVP-I_2_ and AV-PVP-I_2_-NaI are observed at 290 and 359 nm (Figure 4a,c) [29]. These bands are not available in the UV-vis-spectrum of AV-PVP-NaI (Figure 4b). AV absorptions are in the regions from 200–235 nm in the UV-vis spectra of all samples as reported by previous groups (Figure 4) [68,69]. Our AV species is confirmed by the FT-IR spectrum as *Aloe barbadensis* Miller, and is in agreement with previously reported works (Figure 4b) [68]. The presence of PVP is verified by the absorption signal around 220 nm, which is attributed in the literature to originate from the carbonyl groups of polyvinylpyrrolidone [70]. The UV-vis spectra affirm the composition of our biocomplex compounds by the presence of all absorption signals related to AV, PVP and iodine.

The UV-vis spectrum in Figure 4a shows a shift to shorter wavelengths and higher absorption, after adding NaI in the range between 200 and 220 nm, indicating the successful incorporation of NaI into the polymer complex. Wavelength and absorbance are directly correlated to interaction, size and number of surrounding molecules. AV-PVP-NaI shows only absorptions in the range from 200–240 nm (Figure 4b). At the same time, a shift to higher energy in the same range until 240 nm is seen in Figure 4b, when adding PVP and NaI to AV. This confirms stronger interaction and successful incorporation of PVP and NaI into AV. The bands related to iodine and triiodide are missing in AV-PVP-NaI, which confirms its purity.

The size of the complexes increases, starting with AV-PVP-I_2_-Na as the smallest composite, followed by AV-PVP-I_2_ and finally AV-PVP-NaI as the biggest complex compound (Figure 4c). The main components of AV are available in AV-PVP-I_2_ and AV-PVP-I_2_-Na around 290 and 360 nm compared to previously reported data (Figure 4c) [33,68]. According to Moulay [33], two absorption bands of PVP-I_2_ at 305 and 380 nm indicate the adherence of iodine to amorphous regions.

#### 3.1.4. Fourier-Transform Infrared Spectroscopy (FT-IR)

FT-IR analysis of AV-PVP-I_2_ and AV-PVP-I_2_-Na confirmed the composition of the samples. Figure 5 shows the FT-IR spectra of AV-PVP-I_2_ and AV-PVP-I_2_-Na.

The FT-IR spectra of AV-PVP-I_2_ and AV-PVP-I_2_-Na are both similar. The broad band around 3000 to 3600 cm^−1^ is due to OH (alcohol group) and NH (amide group) stretching vibrations (Figure 5). These broad bands with broad bands around 3220 and 3397 cm^−1^ originate from AV and PVP. Asymmetric and symmetric stretching vibrations of methylene (CH_2_) are available for AV-PVP-I_2_ at 2952, 2919 and 2837 cm^−1^ and for AV-PVP-I_2_-Na at 2952, 2920, 2843, and 2864 cm^−1^, respectively (Figure 5). The asymmetric COO-stretching vibrations are present at 1650 and 1648 cm^−1^ for AV-PVP-I_2_ and AV-PVP-I_2_-Na, respectively (Figure 5). Symmetric COO-stretching of the carboxylate groups appear 1403 cm^−1^ for AV-PVP-I_2_ and at 1406 and 1395 cm^−1^ for AV-PVP-I_2_-Na (Figure 5).

The band at 1245 cm^−1^ for AV-PVP-I_2_ and the two bands for AV-PVP-I_2_-Na at 1250 and 1225 cm^−1^ belong to the C–O-C stretching vibrations bands for acemannan, other functional groups of AC bioactive compounds, and are a result of reactions during the encapsulation of AV with PVP (Figure 5). The strong bands in both composites at 1013 cm^−1^ are related to the C-C stretching and CH_2_ rocking vibration of the PVP backbone. The signals around 900 originate from C-C and C-O vibrational stretching in PVP, according to previous reports [67,70]. C-H out of plane deformation vibrations are available from 812–850 cm^−1^ (Figure 5). The bands at 584 cm^−1^ for both composites arise from out-of-plane deformation vibrations of amide groups N-C=O in PVP, according to our previous work [38] and other reports (Figure 5) [67].

### 3.2. Antimicrobial Testing of AV-PVP-I_2_ and AV-PVP-I_2_-Na

#### 3.2.1. Investigation of Antimicrobial Activities of AV-PVP-I_2_ and AV-PVP-I_2_-Na

The composites AV-PVP-I_2_ and AV-PVP-I_2_-Na were tested by disc diffusion assays against 5 Gram-positive, 4 Gram-negative bacterial reference strains and the fungus (Table 1). Five antibiotics were utilized as positive controls (Table 1). The antimicrobial testing was done by dilution of the biohybrids to 50 µg/mL (1), 25 µg/mL (2), and 12.5 µg/mL (3) and 6 µg/mL (4). There are no significant changes in the dilution series, and all the results are in expected ranges. Methanol and water were used as negative controls showing no zone of inhibition (ZOI). The microorganisms were all resistant towards AV-PVP-NaI, AV and AV-PVP. Results with no inhibition zones during the antimicrobial testing were not included in Table 1 and in other following tables.

The bioactive compounds AV-PVP-I_2_ and AV-PVP-I_2_-NaI illustrated evidently vigorous antifungal properties (Table 1, Figure 6). AV-PVP-I_2_ inhibited *C. albicans WDCM 00054* more than AV-PVP-I_2_-NaI (Table 1, Figure 6g,h). Gram-negative and Gram-positive pathogens are also inhibited by both complexes. *P. mirabilis ATCC 29906* is the only resistant microorganism against our two compounds (Table 1). The antimicrobial tests for AV-PVP-I_2_ showed larger ZOI compared to AV-PVP-I_2_-NaI (Table 1, Figure 6). Both compounds exhibited high inhibition zones against *S. aureus ATCC 25923* and the spore forming bacteria *B. subtilis WDCM 00003* at concentrations of 50 µg/mL (Table 1, Figure 6).

#### 3.2.2. Antimicrobial Activities of PGA Sutures Dip-Coated with AV-PVP-I_2_, AV-PVP-I_2_-NaI and AV-PVP-NaI

PGA sutures were impregnated with our three biocomplexes AV-PVP-I_2_, AV-PVP-I_2_-NaI and AV-PVP-NaI. PGA sutures with a length of 2.5 cm were first treated with acetone. The sutures were dried at room temperature for 2 h and immersed into the bioactive compounds with a concentration of 50 µg/mL for 18 h. The diffusion assay on 10 reference strains was performed with PGA sutures 30 min (W), after coating with the biohybrids and other PGA sutures after 3 h drying (D) at ambient temperature (Table 2). The five antibiotics chloramphenicol (C), gentamycin (G), cefotaxime (CTX), strepromycin (S), and nystatin (NY) were utilized as positive controls (Table 2). All microorganisms were resistant against the D and W of AV-PVP-NaI and were excluded from Table 2.

The impregnated PGA sutures with AV-PVP-I_2_ and AV-PVP-I_2_-NaI inhibited all selected pathogens except *P. mirabilis ATCC 29906* and *P. aeruginosa WDCM 00026* (Table 2). The microbial strains showed higher susceptibility to AV-PVP-I_2_ than AV-PVP-I_2_-NaI, with the exception of *K. pneumoniae WDCM 00097* (Table 2, Figure 7). AV-PVP-I_2_ and AV-PVP-I_2_-NaI exerted the highest antimicrobial effects on *C. albicans WDCM 00054,* followed by *S. aureus ATCC 25923* and *B. subtilis WDCM 00003* (Table 2, Figure 7 and Figure 8).

AV-PVP-I_2_ and AV-PVP-I_2_-NaI exerted the highest antimicrobial effects on *C. albicans WDCM 00054,* followed by *S. aureus ATCC 25923* and *B. subtilis WDCM 00003* (Table 2, Figure 7 and Figure 8).

## 4. Discussion

Antibiotic resistance is a danger for the existence of mankind causing morbidity and mortality [1]. Nosocomial infections due to ESKAPE pathogens are a burden on quality of life and health care systems [1,2,3,4]. New antimicrobial compounds are needed to assist the fight against resistant microorganisms. Plant-based antimicrobial agents have promising properties and depend on synergistic strategies [40]. We encapsulated the well-known plant *Aloe vera* and the microbicide iodine in a “smart” biocomplex with PVP, and investigated the antimicrobial properties. The morphology and composition of the biocomplexes were confirmed by SEM and EDS (Figure 1 and Figure 2). The SEM analysis revealed irregular and granular morphology with various particle sizes (Figure 1). The impregnated PGA suture exhibits a firm, homogenous coating on the surface compared to the plain control suture (Figure 1b,c). The EDS and EDS-layered images in Figure 2 verify the composition and uniform distribution of the biocomplexes. AV-PVP-I_2_ consists of C, O, N, I, Cl, K, Na, Mg and Al, with 60.3, 17.4, 8.8, 6.4, 3.5, 2.5, 0.9, 0.1 and 0.1 weight%, respectively (Figure 2a). The sample AV-PVP-I_2_-NaI shows higher weight percentages for Na with 30.3 and I with 2.0 compared to AV-PVP-I_2_ with 0.9 wt% for Na and 6.4 wt% for I (Figure 2c). Ray et al. [52] investigated the EDS of *Aloe vera barbadensis* Miller, and reported Na, Mg, K, Ca, Cd, Al and P, with the two latter minerals in negligible amounts. Rana et al. found in their EDX analysis Al, K, Ca, Si, P and Cu in decreasing weight percentages [71]. The elemental distribution in AV depends highly on soil, location, climate and harvest season [52]. Therefore, the composition may vary individually from region to region and needs standardization for pharmaceutical uses [52].

Figure 1d confirms homogeneity and firm adherence of the biomaterial AV-PVP-I_2_-NaI on the breaded PGA suture. In our previous works, we also used PGA sutures as antimicrobial agents to prevent surgical site infections [38]. The physical adsorption between AV-PVP-I_2_ and AV-PVP-I_2_-NaI biocomposite molecules and the PGA suture material occurs by attractive forces. These attractive forces include Van der Waals forces, with electrostatic attractive forces between uncharged moieties (weak London dispersive forces) and strong dipol-dipol forces, as well as hydrogen bonding [38,72]. AV is composed of many constituents, including phenolic compounds, flavonoids, polysaccharides, carbohydrate polymers and anthraquinones. All these bioactive polymers consist mainly of conjugate aromatic systems with hydrophobic benzene rings and polar functional hydroxyl-, carbonyl-, ether- and acetyl- groups. The absorbable polymer PGA contains many electronegative oxygen atoms in form of ether and carbonyl groups within the polymer chain. PVP has polar oxygen atoms on the amide groups of the positively polarized pyrrolidone rings and is connected by hydrophobic alkyl-backbone. The abundant OH groups in the AV biopolymer form hydrogen bonding with the oxygen atoms in the ether and carbonyl groups available in PGA and amide groups in PVP. The nonpolar entities of PVP and AV constituents physically adsorb through dipolar attraction forces to the polar oxygen atoms within PGA. The nonpolar aromatic groups in AV are also attracted by electrostatic interaction with the polar oxygen groups in the carbonyl entities of PVP [70]. NaI forms ionic bonds through electrostatic attraction to charged moieties, due to loss of conjugation, hydrogen bonding or loss of hydrogen and complexes with PVP [33]. The triiodide ion is formed by combination of molecular iodine and iodide and complexed by PVP [22,33,73,74]. The triiodide ion is electrostatically attracted to polar groups along the PVP backbone, as in PVP-I_2_ [33]. One of the main constituents of AV leaf gel is acemannan. This polysaccharide consists of a mannose backbone, which is intercepted by glucose [56]. The two monosaccharides are linked through β-(1-4) glycosidic bonds and have acetyl groups on the monosaccaride units [56]. The acetyl groups are important for the bioactivity of acemannan [56].

The XRD analysis of the bioactive compounds AV-PVP-I_2_-NaI and AV-PVP-I_2_-NaI affirmed a mainly amorphous nature, in accordance with the results of Ray et al. [52]. Treatment of the AV gel with different mechanical, thermal and drying methods destroys the original structure of the plant, breaks the plant cell walls, leads to degradation, deacetylation and loss of pharmaceutical value [52,56]. We avoided heat and any drying process during the extraction of the AV gel to minimize degradation and deacetylation. Processing the AV gel in a blender, extraction-process and filtration change the previous compact structure into an amorphous liquid. The XRD analysis of AV-PVP-I_2_ reveals a semicrystalline nature of the sample due to two sharp peaks (2θ = 9° and 14°) and broad signals (around 2θ = 19°, 21.5° and 62.5°). The peaks at 9°, 14° and 21.5° are related to the biomolecules within *Aloe vera Barbadensis* Miller, corresponding to reflexions from the planes (110) and (220) (Figure 3a) [64,65]. The broad signal at 19° indicating amorphous nature is due to PVP according to XRD results of previous groups [67,75]. The biocomplex AV-PVP-I_2_-NaI reveals a change for the AV biocomponents from crystalline to amorphous nature due to the additional NaI (Figure 3b). New sharp peaks arise in AV-PVP-I_2_-NaI at 2θ = 27° and 40.5° confirming a crystalline nature of NaI (Figure 3b). There is also a hump at 2θ = 59° - 67° in both samples, which is related to iodine [38], or as previously reported to organometallic materials [76]. The data are in agreement with the model of intercalation of iodine between planes of close-packed polymer chains [77].

The UV-vis spectra of AV, AV-PVP-I_2_, AV-PVP-I_2_-NaI, and AV-PVP-NaI are shown in Figure 4. AV-PVP-NaI shows only absorptions in the region between 200 and 225 nm, which are related to PVP and AV (Figure 4b). PVP absorbs at 220 nm due to the carbonyl groups, and in agreement with previous reports and the signals are seen in all the three UV-spectra of our biohybrids (Figure 4) [70]. The carbonyl groups of PVP absorb at AV-PVP-I_2_, AV-PVP-I_2_-NaI, and AV-PVP-NaI.

Wei at al. reported iodine absorption at 203 nm, triiodide at 288 and 352 nm, and iodide at 193 and 226 nm [78]. Li et al. observed the absorption signal for iodine at 460 nm, the triiodides appeared at 297 and 350 nm, while the iodine was confirmed at 293 nm [79]. The cyclodextrin-triiodide complex showed a band for iodine at 460 nm; two absorption bands due to triiodide ions at 290 and 350 nm, and iodide absorption at 192 and 225 nm [73]. Another group mentioned for their cyclodextrin-pentaiodide complex signals for iodine at 205 and 460 nm [74]. The same group also observed absorption bands for the triiodide ion at 290 and 352 nm [74]. These reports denote the expected absorption ranges for iodine, triiodide and iodide, and can be used to verify our results (Table 3).

In the UV-vis spectrum of our sample AV-PVP-NaI, there is a small shoulder at 226 nm, indicating the presence of iodide ions, but no signals related to molecular iodine, nor triiodides appear (Figure 4b). This confirms the composition of AV-PVP-NaI.

In previous reports, molecular iodine absorbs at around 203 and 460 nm. In the UV-vis spectra of our complexes AV-PVP-I_2_ and AV-PVP-I_2_-NaI, there is an absorption around 203 nm, but no additional absorption at 460 nm. The hydrolysis of molecular iodine produces triiodide ions:[I-I] + I^−^ ⇆ [I-I^…..^I]^−^(1)
thereby dramatically reducing the absorption bands of molecular iodine [74]. At the same time, the triiodide ions are adsorbed to the PVP by hydrogen bonding to the carbonyl groups of the polymer PVP, forming PVP-I_3_^−^ [33]. PVP-I_3_^−^, which is referred to as PVP-I_2_, is incorporated into the amorphous regions of the polymer PVP, giving rise to absorbance at 310, 361 and 395 nm. The highest absorbtion is at λ_max_ = 361 nm [33]. In our samples AV-PVP-I_2_ and AV-PVP-I_2_-NaI, there is a clear shoulder at 305 nm, verifying the presence of PVP-I_3_^−^ in the form of a PVP-[I-I-I]^−^ complex (Figure 9).

The spectra of AV-PVP-I_2_ and AV-PVP-I_2_-NaI show broad absorption signals due to triiodide ions around 290 and 359 nm (Figure 4a,c) [22,29]. In comparison to above host-guest inclusion cyclodextrin complexes with absorptions for the triiodide ions at 350 and 352 nm, our biohybrids AV-PVP-I_2_ and AV-PVP-NaI have a red shift towards 359 nm [73,74]. This bathochromic effect underlines a change in bond distances compared to the two cyclodextrin-complexes, and especially to pure triiodide ions:AV-PVP + [I-I^…..^I]^−^ ⇆ AV-PVP-[I-I-I]^−^(2)
which absorb at 359 nm [73,74].

The broad signals from 286–296 and 353–363 nm suggest joint absorption of AV components and triiodides in this region. At the same time, we suggest that two types of triiodides exist within our samples, consisting of free [I-I^…..^I]^−^ and complexed [I-I-I]^−^, as denoted in Equation (2). The concentration of complexed triiodide ions is more than the free iodide ions, therefore, the absorption band λ_max_ is at 359 nm. The previously reported “smart” triiodide compound shows the same absorption signals at 290 and 359 nm, and this confirms pure halogen bonding within the triiodide units [22,29]. The complexation with AV-PVP produces AV-PVP-[I-I-I]^−^ complexes within our biomaterials AV-PVP-I_2_ and AV-PVP-I_2_-NaI. The I_3_^−^–groups absorptions in the UV-Vis spectrum are characteristic of pure halogen bonding within “smart” triiodide units [22,29]. The anionic structure is comprised of isolated, symmetrical triiodide units [I-I-I]^−^ with halogen bonding. Halogen bonding in triiodides increases the antimicrobial properties due to the controlled release of molecular iodine upon deformation of the host-complex compound when electrostatically interacting with the cell membrane molecules of pathogens [22].

In Figure 4c, a comparison between all four UV-vis spectra easily reveals a general trend. A hypsochromic shift towards shorter wavelengths starting from the spectrum of AV, to AV-PVP-I_2_, to AV-PVP-NaI, and finally AV-PVP-I_2_-NaI is evident (Figure 4c). This blue shift verifies increasing molecular interaction, successful incorporation of the added components (AV/I_2_/NaI) into the PVP-polymer backbone, and decreasing size of the complexes towards the system AV-PVP-I_2_-NaI. These findings affirm the complexation of the added moieties by the polymer backbone, and the resulting influence on the mobility of the whole molecular structure of the biocomlexes, as well as the reduction of the chromophores by hydrogen bonding. The incorporation of these moieties is due to a decrease in conjugation as a result of hydrogen bonding. The number of π-electrons are reduced by saturating -C=O groups of PVP through hydrogen bonding, with hydroxyl groups of the AV components. Other reasons are the deacetylation of acemannan and the degradation of few bioactive components of AV [49,81]. The same general trend takes place with the same reasons when the absorption intensity of AV-PVP-I_2_ and AV-PVP-I_2_-NaI are compared. AV-PVP-I_2_-NaI has a higher absorption intensity compared to AV-PVP-I_2_. This hypochromic effect is related to the addition of the ionic compound NaI into AV-PVP-I_2_:I_2_ + NaI ⇆ Na^+^[I-I^…..^I]^−^(3)
AV-PVP-[I-I-I]^−^ + Na^+^[I-I^…..^I]^−^ ⇆ AV-PVP-[I-I-I]^−^ Na^+^ + [I-I^…..^I]^−^(4)
which leads to more free triiodide ions in the solution.

The hydrophobic aromatic conjugate systems originating from AV are already incorporated into the PVP-backbone, together with triiodide ions in AV-PVP-I_2_. The hydrophilic, ionic compound NaI causes additional disturbance by repulsion, increasing steric crowding and steric hindrance, which results in the desorption of different phenolic compounds and triiodides [38]. The high absorption intensities at 290 and 359 nm in both biocomplexes are due to high numbers of triiodide ions within the complex AV-PVP-[I-I-I]^−^ and free triiodides in solution [73,74].

Adding sodium iodide increases the concentration of iodide, which leads to the production of triiodide, reducing the molecular iodine concentration, and competing with the encapsulated AV biomaterials on the PVP-backbone. Accordingly, AV-PVP-I_2_ retains the AV-aromatic compounds more effectively than AV-PVP-I_2_-NaI, and can be considered as the better biomaterial with higher antimicrobial potential. The absorption signals around 200–215 nm, 290 nm and 359 nm affirm the availability of triiodide, iodide and iodine in the PVP-polymer backbone.

The UV-vis analysis of the pure AV extract is shown in Figure 4b. The abundance of absorption signals and their intensity are a confirmation for the successful extraction of many bioactive compounds from our AV leaves. Our extraction method kept most of the active ingredients like anthraquinones, flavonoids and phenolic compounds intact during the treatment, and this resulted on the UV analysis. The findings of the prominent components in our methanolic extract AV are listed in Table 4.

The main components of AV are aloin, aloe-emodine, aloesin, aloesin derivates, rhein and phenolic compounds pyrogallol and hesperidine, according to previous reports (Table 4) [50,68,69,82,83,84,85]. The flavonoids within AV mainly absorb in the UV-B region and the UV-vis spectrum of [86]. In the UV-A region of the spectrum between 335 and 375 nm, phenolic compounds within AV are usually showing UV-vis absorption signals [86]. UV-vis absorption of anthraquinones and polysaccharides of AV can be expected in the UV-C region. Our bioactive polymers AV-PVP-I_2_-NaI and AV-PVP-NaI show absorption signals throughout these regions. This is a clear indication for the successful encapsulation of AV biomolecules and triiodide ions into the PVP-backbone in our biocomplexes. The UV-vis spectra verify the composition of our four samples AV, AV-PVP-I_2_, AV-PVP-NaI, and AV-PVP-I_2_-NaI.

The FT-IR spectra show the characteristic signals originating from the bioactive compounds, PVP and Na (Figure 5). The AV components verified by FT-IR are acemannan, aloin, aloe-emodine, aloesin, mannose and glucan (Figure 5). The FT-IR spectrum of pristine AV methanolic extract, PVP-I_2_ (both as Appendix A) compared with literature data, UV-vis results and FT-IR spectra confirmed the composition of our biocomposites AV-PVP-I_2_ and AV-PVP-I_2_-NaI (Figure 5). Both samples show a similar pattern in the FT-IR spectra (Figure 5).

The stretching vibrations of hydroxyl (alcohol group) and NH (amide group) in AV and PVP appear as broad band around 3000 to 3600 cm^−1^ (Figure 5). A sharp band in the spectrum of AV-PVP-I_2_-NaI at 3669 cm^−1^ denotes OH stretching in PVP due to its hydrophilic nature, as well as free hydroxyl groups from carbohydrate monomers mannose, galacturonic acid, and uronic acid [70,81]. The broad bands around 3220, 3228 and 3397 cm^−1^ are related to PVP and AV [81]. Compared to the pure AV methanolic extract, there is a reduction in the size of these broad bands for hydroxyl groups (Appendix A). This can be related to hydrogen bonding and reactions between hydroxyl groups in AV components with the carbonyl C=O of PVP. The oxidation of AV components like aloin and further anthraquinones is also possible, supported by the almost missing C=C stretching related signals around 1600 cm^−1^ [81]. The asymmetric and symmetric stretching vibrations of methylene (CH_2_) for AV-PVP-I_2_ appear at 2952, 2919 and 2837 cm^−1^ and for AV-PVP-I_2_-Na at 2952, 2920, 2843, and 2864 cm^−1^, respectively (Figure 5). According to previous reports, the signals at 2952 cm^−1^ indicate the presence of aloin and PVP through its asymmetric methyl group stretching vibration [50,67]. The weak signals at and around 1919 cm^−1^ are related to symmetric C-H-stretching vibrations of chains due to the alkyl polymer chain in PVP and are attributed to the polysaccharide acemannan [67,87]. The hydroxyl groups on methylene moieties appear at 2837 and 2843 for AV-PVP-I_2_ and AV-PVP-I_2_-Na, respectively [88]. The vibrational bands for the CH_2_-deformation on the alkyl chain of PVP are available at 2359 and 2334 in both samples (Figure 5) [67,70]. The band at 2146 cm^−1^ belongs to the C=O combination vibrations in PVP [70]. The asymmetric COO-stretching vibrations originating from AV components and PVP are present at 1650 and 1648 cm^−1^ for AV-PVP-I_2_ and AV-PVP-I_2_-Na, respectively (Figure 5) [67,69,70,81,87,88]. The related AV components include acemannan and aloin (Figure 5) [50,87,89]. The pure AV methanolic extract reveals strong and broad signals around 1700 cm^−1^, while the complexed biomaterials show only a medium to weak sized, almost sharp signal at 1640 cm^−1^ [87]. This reduction of signals, together with a weak C-O-C stretching vibration at 1245 cm^−1^, indicates a degree of deacetylation in acemannan [87]. The degree of acetylation is an indicator for enhanced interaction with other bioactive compounds, increased hydrophilicity and hydrogen bonding abilities. The acetate groups are attacked by nucleophiles on the partial positively charged carbon atom acetyl group, and substituted by removing the acetyl-group [87]. Deacetylation occurs during extraction and drying processes, reduces steric hindrance and increases hydrogen bonding [56,81,87]. Drying processes like industrial freeze drying, spray drying, radiant zone drying or even coating lead to deacetylation in acemannan [56]. In our work, we tried to extract without any drying process to preserve acetylation of acemannan. This method was effective in preserving acemannan in our AV sample and partly in the two biocomplexes AV-PVP-I_2_, AV-PVP-I_2_-Na, as confirmed by the FT-IR and UV-vis analysis. We also referred to this phenomenon in the discussion of the UV-Vis-spectra above. The location and reduced size of the C=O vibration signal at 1650 cm^−1^ strongly indicate reactions between carbonyl oxygen atoms in C=O of pyrrolidone rings in PVP. These reactions include hydrogen bonding with methanol, hydroxyl groups on AV compounds, as well as ionic bonds with sodium ions [33,81]. The most important hydrogen bonding mechanism is due to a hydrogen bridge, formed between the two carbonyl groups of pyrrolidone rings as it occurs in PVP-I_2_ [33]. This positively charged hydrogen bridge adheres triiodide ions and is the reason for the excellent antimicrobial activity of PVP-I_2_ [33]. We confirmed the availability of this group by the signal around 361 nm and PVP-related absorptions in the FT-IR analysis.

The bands around 1472 cm^−1^ in both samples are related to C-H bending of CH_2_-groups in PVP [75]. Symmetric COO-stretching of the carboxylate groups corresponding to bioactive components in AV appear at 1403 cm^−1^ for AV-PVP-I_2_ and at 1406 for AV-PVP-I_2_-Na (Figure 5) [64,81,88]. The weak signal around 1395 cm^−1^ denotes the presence of aloin [50]. The in-plane C-H bending of H-C-OH glycoside units in acemannan appear at around 1361 cm^−1^ (Figure 5) [50,87]. The C-N stretching and CH_2_ wagging in PVP are represented by the weak bands at 1291 cm^−1^ [67,75].

The band at 1245 cm^−1^ for AV-PVP-I_2_ and the two bands for AV-PVP-I_2_-Na at 1250 and 1225 cm^−1^ originate from the C–O-C stretching vibrations bands of acemannan and further AV components (Figure 5) [64,75,87,89]. These absorptions are also due to reactions between C=O groups of PVP and hydroxyl groups in AV during the encapsulation process, and between carbonyl groups in AV with methanol during extraction (Figure 10).

C-O stretching and C-O aliphatic amine vibrations from the pyranose ring, acemannan, and monopyranose are indicated by the weak signal at 1066 cm^−1^ [64,81,89,90]. The strong bands in both composites at 1013 cm^−1^ are related to the C-C stretching and CH_2_ rocking vibration of the PVP backbone [67]. The signals around 904, 913, 933, 945 originate from C-C and C-O vibrational stretching in PVP according to previous reports [67,70]. C-H out of plane deformation vibrations due to PVP, mannose and pyranoside ring are available from 812-860 cm^−1^ (Figure 5) [67,81,82,88,90]. Weak absorption bands around 661 cm^−1^ indicate the presence of C-H bending in polymerized carbohydrates and phenolics [53]. The bands around 584 cm^−1^ for both composites arise from out-of-plane deformation vibrations of amide groups N-C=O in PVP (Figure 5) [38,67,82]. In general, the FT-IR spectra of the two biocomposites AV-PVP-I_2_, AV-PVP-I_2_-Na are similar, the above explanations related to occurring signals are valid for both samples. The difference between the two FT-IR spectra is the higher intensity of the sample AV-PVP-I_2_ compared to AV-PVP-I_2_-Na in all the FT-IR measurement, except around the region for the OH-vibration. This indicates higher acetylation degree, less degradation and oxidation of acemannan, aloin and further compounds. At the same time, it verifies the higher degree of encapsulation of AV by PVP in AV-PVP-I_2_. The addition of NaI in AV-PVP-I_2_-Na caused disturbances in the encapsulation by increased repulsion, removal of AV compounds from the PVP backbone and steric crowding. The reduced encapsulation by PVP resulted in oxidation of AV component and deacetylation [56,60]. The degree of acetylation is a direct marker for antimicrobial activity and the decrease in acetylation in AV-PVP-I_2_-Na means less antimicrobial activity.

There are no significant changes in the dilution series, and all the results are in expected ranges. All selected microorganisms were resistant towards AV, AV-PVP-NaI, and AV-PVP. These samples showed no antimicrobial activities towards the selected microorganisms in our disc dilution assays, because they do not contain molecular iodine. At the same time, AV-PVP-NaI shows no activity for any of its components towards all pathogens. This affirms that the iodide ions, the sodium ions, the AV biohybrids and PVP within this biocomplex do not play any role in the inhibition of the pathogens. After adding molecular iodine, the antimicrobial action arises through synergism with the other components within the encapsulated complexes AV-PVP-I_2_ and AV-PVP-I_2_-NaI. The UV spectra verified the presence of triiodide ions in these two biomaterials. Triiodides are the key for free molecular iodine release and subsequent inhibition of pathogens [22,29]. At the same time, we can verify that the compound AV-PVP-NaI does not lead to the formation of I_2_ molecules. We were not able to detect.

The biocomplexes AV-PVP-I_2_ and AV-PVP-I_2_-NaI proved to have strong antifungal properties in comparison to the control antibiotic nystatin (Table 1, Figure 6). In comparison, AV-PVP-I_2_ inhibited *C. albicans WDCM 00054* more than AV-PVP-I_2_-NaI, but the later kept inhibiting until a dilution of 3 µg/mL (Table 1, Figure 6). This proves that AV-PVP-I_2_ initiated overwhelming strong inhibition immediately upon contact at 50 µg/mL with a ZOI = 56 mm against *C. albicans* and reduces slowly the action during further dilution. AV-PVP-I_2_-NaI exerts longer lasting inhibition during further dilutions compared to AV-PVP-I_2_. AV-PVP-I_2_ shows an immediate, strong action at 50 µg/mL with an inhibition zone of ZOI = 35 mm, but continues to inhibit until concentrations of 3 µg/mL with a ZOI = 9 mm. AV-PVP-I_2_-NaI exerts longer antimicrobial activity slowly and steadily in a more controlled manner until the lowest concentrations, when compared to AV-PVP-I_2_ (Figure 11).

The longer lasting inhibition of AV-PVP-I_2_-NaI compared to AV-PVP-I_2_ is due to the higher amount of triiodide ions in the biocomplexes, as shown in Figure 9.

*C. albicans WDCM 00054* is highly susceptible to our biocomposites AV-PVP-I_2_ and AV-PVP-I_2_-NaI. The next highly inhibited microorganisms are the Gram-positive pathogen *S. aureus ATCC 25923* and *Bacillus subtilis WDCM 00003*. AV-PVP-I_2_ and AV-PVP-I_2_-NaI showed against *S. aureus ATCC 25923* inhibition zones of 25 and 20 mm, and *Bacillus subtilis WDCM 00003* 22 and 16 mm, respectively. Gram-positive pathogens *E. faecalis ATCC 29212*, *S. pyogenes ATCC 19615* and *S. pneumoniae ATCC 49619* were inhibited by AV-PVP-I_2_ with ZOI = 17, 16 and 14 mm, respectively. AV-PVP-I_2_-NaI inhibited *S. pneumoniae ATCC 49619* and *S. pyogenes ATCC 19615* with ZOI of 13 and 10 mm. Fungi and Gram-positive bacteria have no porin channels, which could have been responsible for the antimicrobial action of iodide and triiodide [91,92,93]. These two moieties are small, charged and lipohilic [22]. Due to their negative charges, they cannot pass through the lipophilic inner parts of the cell membranes of microorganisms. This verifies again, that these two species cannot be the reason for antimicrobial activity in the yeast *C. albicans WDCM 00054* and all other Gram-positive bacteria. The reason for the high antimicrobial activity against these two pathogens and all the other selected microorganisms in our two samples is the release of molecular iodine from the triiodide units, which are adhered to the PVP backbone [33,91]. The release of free molecular iodine is due to deformations of the structure caused by interactions of PVP with the pathogen cell membranes. The interactions happen between partly positively charged the pyrrolidone ring carbon- and nitrogen atoms and alkyl chain carbon atoms in the PVP with negatively charged cell membrane moieties of the cell membranes in microorganisms. Additionally, in Gram-positive bacteria, these include the oxygen atoms in the peptide interbridges of the thick peptidoglycan layer and the negatively charged peptidoglycans with their teichoic and lipoteichoic acid inclusions [22,91,92]. In Gram-negative pathogens, further sites of negatively charged molecules are the outer leaflets of the cell membrane with their lipopolysaccharides. The electrostatic interactions between partial negatively charged groups in pathogens with the partial positively charged atoms in the PVP-backbone cause the release of free molecular iodine, which lead to microbial membrane damage. According to our UV-vis analysis, the complexed triiodide moieties are in the form of a linear, symmetrical three-center-system [I-I-I]^−^ with strong halogen bonding [22,29]. The free molecular iodine is released after interactions with microbial components-our biohybrids lead to deformations in the PVP-I_3_^−^ structure, which change the linear, symmetrical three-center-system [I-I-I]^−^ into [I-I···I]^−^ units. The same mechanism happened in our previous work with the triiodide complex [Na(12-crown-4)_2_]I_3_ [22]. PVP cannot pass through the bacterial cell membranes like chitosan, due to its size and high molecular weight [22,92,94,95]. Instead, our PVP-biocomplexes are attracted through electrostatic interactions to the cell membranes of microorganisms like [Na(12-crown-4)_2_]I_3_ and chitosan, another high molecular weight natural biopolymer [22,92,94,95]. The electrostatic interactions can result in the hydrolysis of peptidoglycans, iodination of cell membrane molecules and cell membrane permeability changes [22,91,92,93,94]. These mechanisms lead to the leakage of intracellular components, inhibition of bacterial growth, changes in the bacterial metabolism and finally cell death [22,91,92,93,94]. As a result, the antimicrobial action of our two bioactive complexes depends on the release of free iodine from the symmetrical, and energetically more stable triiodide units adhered to the PVP backbone through hydrogen interbridges between the carbonyl atoms of the pyrollidone rings. Free molecular iodine passes through the microbial cell membranes and cause intracytoplasmic protein oxidation [22,91,96]. Solvents play an important role in the extraction of the bioactive compounds in AV. Cheng et al. [97] observed that aloe-emodin and further anthraquinone derivates coexisting with aloe emodin were more soluble in ethanol and methanol compared to water. Stanley et al. [98] extracted AV gel with methanol and tested the filtrates against *E. coli*, *S. aureus* and *C. albicans*. Only *E. coli* was inhibited by the AV extract, while the other microorganisms were resistant [98].

AV-PVP-I_2_ and AV-PVP-I_2_-NaI inhibited the Gram-negative bacteria *K. pneumoniae WDCM 00097* (ZOI = 23 and 11 mm), *E. coli WDCM 00013* (ZOI = 18 and 12 mm), *P. aeruginosa WDCM 00026* (ZOI = 13 and 9 mm), at concentrations of 50 µg/mL. *P. mirabilis ATCC 29906* proved to be resistant against all our compounds (Table 1). In general, our selection of pathogens were more susceptible towards AV-PVP-I_2_ than AV-PVP-I_2_-NaI. AV-PVP-I_2_ had higher antimicrobial activities compared to AV-PVP-I_2_-NaI. Our biocomplexes showed remarkable antifungal properties against *C. albicans WDCM 00054*.

PGA sutures were coated with AV-PVP-I_2_ and AV-PVP-I_2_-NaI, tested by disc dilution assay against the same selection of 10 microorganisms, and compared with the antibiotics as positive control (Table 2). Both biocompounds showed against *C. albicans WDCM 00054* the highest inhibition zones and proved to be excellent antifungal agents on dip-coated surgical sutures. The inhibition zones for AV-PVP-I_2_ were at a concentration of 50 and 25 µg/mL 20 and 9 mm, while AV-PVP-I_2_-NaI achieved 4 and 2 mm, respectively (Table 2). The other coated sutures achieved similar results like the disc diffusion assays. Among the Gram-positive bacteria, *S. aureus ATCC 25923* was most susceptible against AV-PVP-I_2_ and AV-PVP-I_2_-NaI, with ZOI = 7 and 6 mm, respectively. AV-PVP-I_2_ inhibited *Bacillus subtilis WDCM 00003*, *E. faecalis ATCC 29212* and the Gram-negative *E. coli WDCM 00013* with ZOI of 5 mm. The diameters of the inhibition zones for *S. pyogenes ATCC 19615*, *K. pneumoniae WDCM 00097* and *S. pneumoniae ATCC 49619* were recorded as 3, 2.5 and 2 mm, respectively. The biocomplex AV-PVP-I_2_-NaI achieved ZOI of 6, 4, 4 and 1 mm at 50 µg/mL for *S. aureus ATCC 25923*, *K. pneumoniae WDCM 00097*, *Bacillus subtilis WDCM 00003*, and *S. pneumoniae ATCC 49619*, respectively (Table 2).

These results confirm the high antifungal and antibacterial activities of our dip-coated PGA sutures and underline their potential use in wound sites to prevent surgical site infections. AV-PVP-I_2_ exerted in dip-coated sutures higher antimicrobial properties than AV-PVP-I_2_-NaI.

We reported in our previous work about the antimicrobial properties in-vitro and on dip-coated sutures of biosynthesized antimicrobial silver nanoparticles (AgNP) with povidone-iodine (PI) and *Cinnamomun zeylanicum* (Cinn) extracts [38]. The combination of three high potential antimicrobial agents AgNP, PI and Cinn in the form of AgNP-Cinn and AgNP-Cinn-PI showed inhibition zones 14–10 mm on the same strains of pathogens as used in this work, at the same concentration of 50 µg/mL [38]. The phenolic compounds in Cinn mainly consisting of cinnamaldehyde inhibited in synergism with PI and AgNP less than our current biohybrids AV-PVP-I_2_ and AV-PVP-I_2_-NaI. This is a clear sign that there is a much stronger synergism in the components of AV with the PVP and the iodine content. AgNP, commonly used to increase the antimicrobial activities, has also adverse effects on health and environmental when used on a large scale. At the same time, it is an expensive resource. The use of AV instead of AgNP showed better results, even without utilizing the strong commercial antimicrobial PI. The PVP in the present biocomposites successfully incorporated the iodine as “smart”, halogen bonded triiodide units and encapsulated the AV in effective way. We reported in another previous work about “smart” triiodides and proved their stability and excellent antimicrobial properties [22]. The nanocomposites AgNP-Cinn and AgNP-Cinn-PI showed no inhibition of *C. albicans WDCM 00054*, while AV-PVP-I_2_ and AV-PVP-I_2_-NaI proved to be excellent antifungal agents. These results reveal the superiority of AV-PVP-I_2_ and AV-PVP-I_2_-NaI compared to AgNP-Cinn and AgNP-Cinn-PI, in terms of being potential antimicrobial agents and their use in wound infection. Further studies are required in future works to validate this thesis by in vivo experiments.

As a result, AV-PVP-I_2_ and AV-PVP-I_2_-NaI are excellent antifungal agents against *C. albicans WDCM 00054* at concentrations of 6 and 3 μg/mL, respectively (Figure 11). The dip-coated sutures of AV-PVP-I_2_ inhibited the fungus at concentrations of 50 μg/mL with a ZOI of 20 mm (Table 2). Dip-coated sutures of AV-PVP-I_2_ and AV-PVP-I_2_-NaI strongly inhibited *S. aureus ATCC 25923* at a concentration of 50 μg/mL, while *E. coli WDCM 00013*, *K. pneumoniae WDCM 00097* and *Bacillus subtilis WDCM 00003* were inhibited intermediately at the same concentration against AV-PVP-I_2_ (Table 2). In vitro diffusion assays showed strong results until 12.5 μg/mL with ZOI = 19 mm for AV-PVP-I_2_ and intermediate results at 25 μg/mL, with 13 mm for AV-PVP-I_2_-NaI (Table 1).

## 5. Conclusions

Antibiotic resistance is a dangerous phenomenon and is already causing an economic burden on the health care system with increased treatment duration, morbidity and mortality. These problems arise through multidrug resistant microbial strains, biofilm formation, especially in hospital settings and nosocomial infections. Increasing burden on the health care system, treatment failures, prolonged treatment duration, morbidity and mortality are linked to the resistance of pathogens towards some existing drugs and antimicrobial agents. New generations of low-cost plant-iodine-based, biosynthesized agents with high effectiveness and abundant sources are needed. Our biocomplexes AV-PVP-I_2_ and AV-PVP-I_2_-NaI and their dip-coated PGA surgical sutures proved to have excellent antifungal properties against *C. albicans WDCM 00054*. Strong inhibitory effects at concentrations of 50 μg/mL against *S. aureus ATCC 25923* were revealed by disc diffusion assay and impregnated PGA sutures. *Bacillus subtilis WDCM 00003*, *E. coli WDCM 00013*, *P. aeruginosa WDCM 00026*, and *K. pneumoniae WDCM 00097* bacterial reference strains were inhibited intermediately. Our biocompounds with PVP encapsulated *aloe vera barbadensis* Miller, and iodine showed higher antimicrobial activity than our previously investigated bio-nanocomposites with silver nanoparticles, povidone iodine and *cinnamomum zeylanicum* extracts, at the same concentration against the same reference strains *in-vitro* and dip-coated on PGA sutures. Further in vivo investigations on our new, cost-effective, non-toxic antimicrobial agents AV-PVP-I_2_ and AV-PVP-I_2_-NaI are needed to undermine the expected antimicrobial action on sutures. Their facile, rapid and low-cost profile opens up a spectrum of potential uses as disinfectants, sanitizers, coating materials in personal protective equipment (PPE), health care settings, public spaces and indoor environments.

## Figures and Tables

**Figure 1 biomimetics-05-00045-f001:**
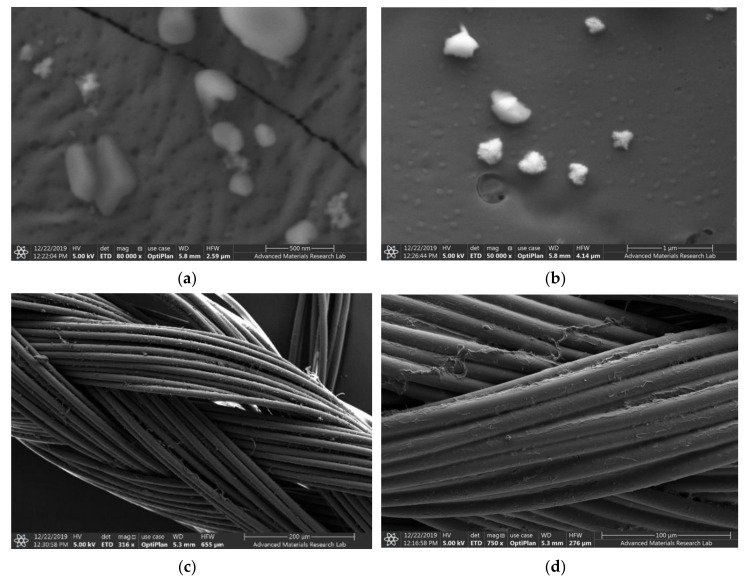
Scanning electron microscopy (SEM) of AV-PVP-I_2_ with *Aloe vera* (AV), polyvinylpyrrolidone (PVP), iodine (I_2_), and sodium iodide (NaI) and the PGA sutures. From up to down: (**a**,**b**) AV-PVP-I_2_-NaI; (**c**) Control (plain PGA suture); (**d**) suture coated with AV-PVP-I_2_-NaI.

**Figure 2 biomimetics-05-00045-f002:**
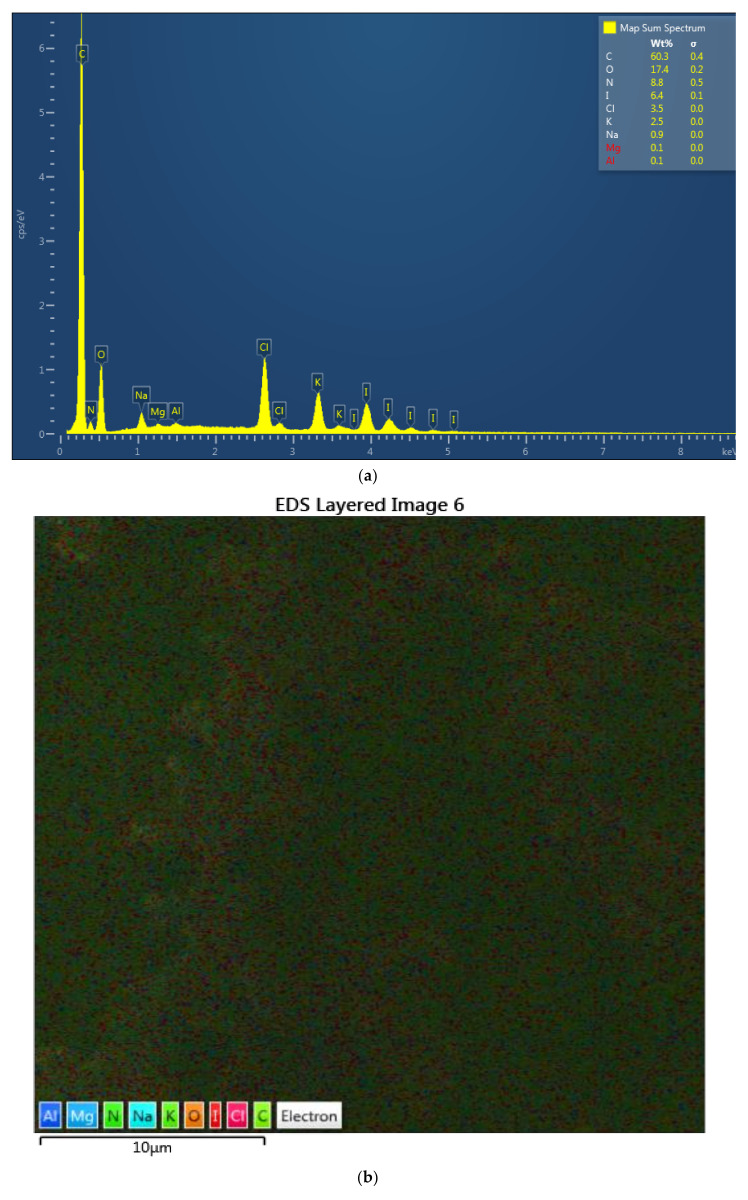
Energy dispersive spectroscopy (EDS) and EDS layered images of the two complexes. From up to down: (**a**) EDS of AV-PVP-I_2_; (**b**) EDS layered image of AV-PVP-I_2_; (**c**) EDS of AV-PVP-I_2_-NaI; (**d**) EDS layered image of AV-PVP-I_2_-NaI.

**Figure 3 biomimetics-05-00045-f003:**
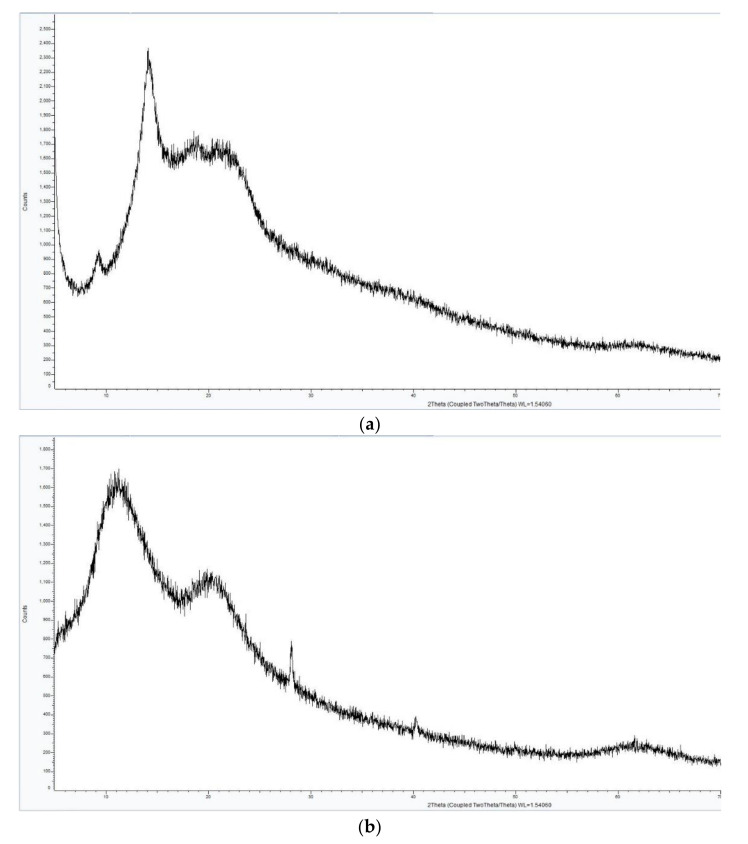
X-ray diffraction (XRD) analysis of two complexes. From up to down: (**a**) AV-PVP-I_2_; (**b**) AV-PVP-I_2_-NaI.

**Figure 4 biomimetics-05-00045-f004:**
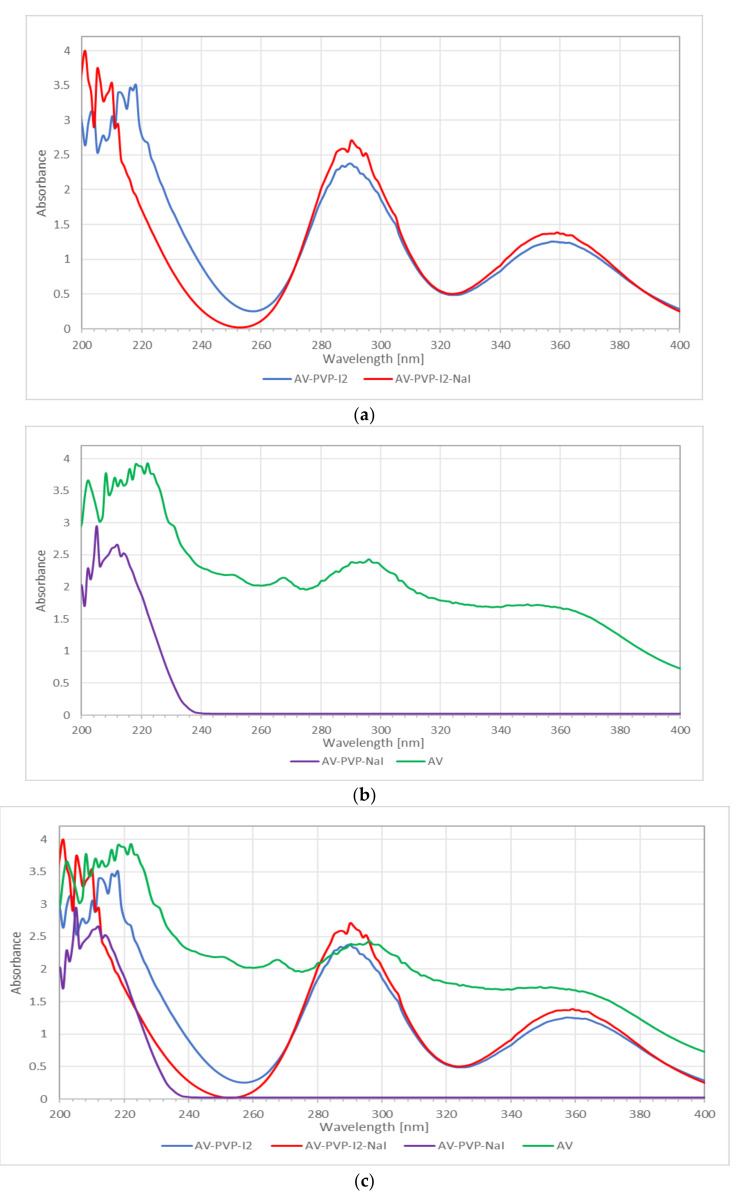
Ultraviolet-visible (UV-vis) spectrometric analysis of the samples. From up to down: (**a**) AV-PVP-I_2_ and AV-PVP-I_2_-NaI; (**b**) AV and AV-PVP-NaI; (**c**) AV, AV-PVP-I_2_ and AV-PVP-I_2_-NaI, and AV-PVP-NaI.

**Figure 5 biomimetics-05-00045-f005:**
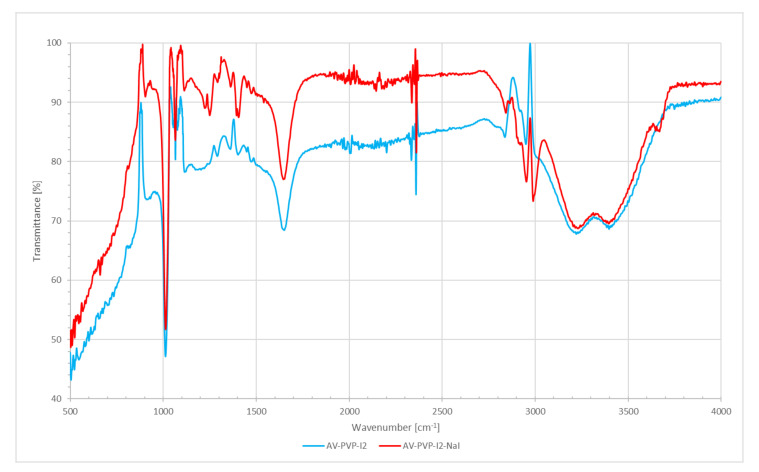
Fourier-transform-infrared (FT-IR) spectrometric analysis of AV-PVP-I_2_ and AV-PVP-I_2_-Na.

**Figure 6 biomimetics-05-00045-f006:**
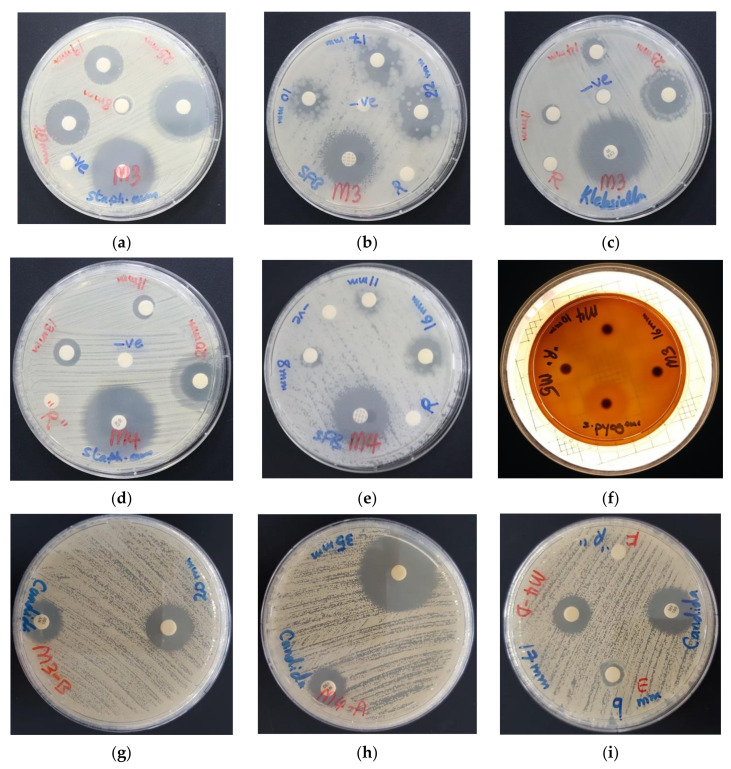
Antimicrobial disc dilution assay of biocomplexes with positive controls (antibiotic). From left to right: AV-PVP-I_2_ against (**a**) *S. aureus ATCC 25923*; (**b**) *P. aeruginosa WDCM 00026;* (**c**) *B. subtilis WDCM 00003*. From (**d**,**e**): AV-PVP-I_2_-NaI against (**d**) *S. aureus ATCC 25932*; (**e**) *B. subtilis WDCM 00003*; (**f**) AV-PVP-I_2_, AV-PVP-NaI and AV-PVP-NaI against *S. pyogenes ATCC 19615*. From (**g**–**i**): Susceptibility of *C. albicans WDCM 00054* towards (**g**) AV-PVP-I_2_ (25 µg/mL); (**h**) AV-PVP-I_2_-NaI (50 µg/mL); (**i**) AV-PVP-I_2_-NaI (D = 6 µg/mL, E = 3 µg/mL, F = 1.5 µg/mL).

**Figure 7 biomimetics-05-00045-f007:**
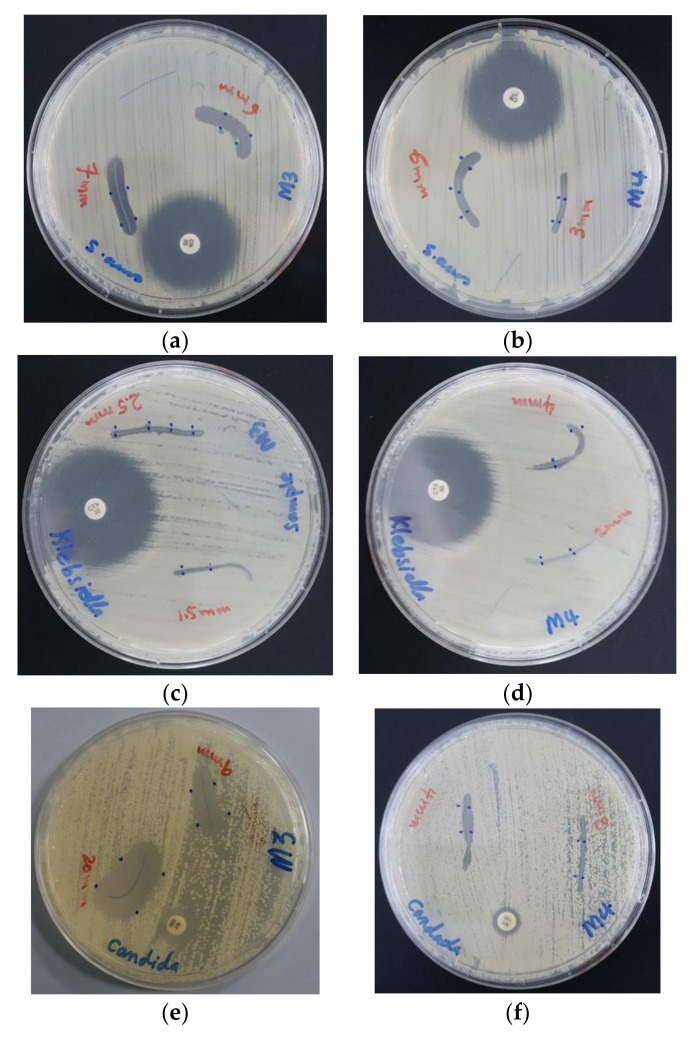
Antimicrobial agar plate methods on dip coated PGA sutures with positive control antibiotics. From left to right: against *S. aureus ATCC 25,923* (**a**) AV-PVP-I_2_; (**b**) AV-PVP-I_2_-NaI; against *K. pneumoniae WDCM 00097* (**c**) AV-PVP-I_2_; (**d**) AV-PVP-I_2_-NaI; against *C. albicans WDCM 00054* (**e**) AV-PVP-I_2_; (**f**) AV-PVP-I_2_-NaI.

**Figure 8 biomimetics-05-00045-f008:**
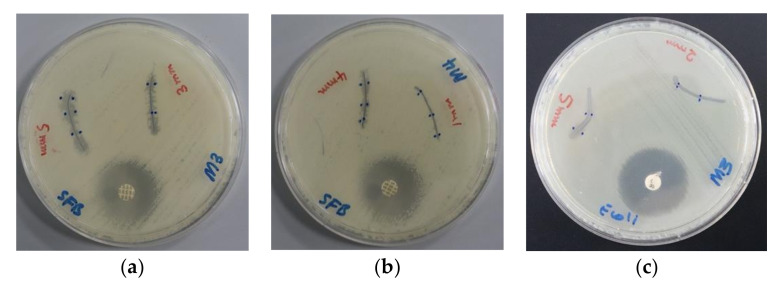
Antimicrobial agar plate methods of dip-coated PGA sutures. From left to right: *Bacillus subtilis WDCM 00003* against (**a**) AV-PVP-I_2_; (**b**) AV-PVP-I_2_-NaI; and *E. coli WDCM 00013* against (**c**) AV-PVP-I_2_.

**Figure 9 biomimetics-05-00045-f009:**
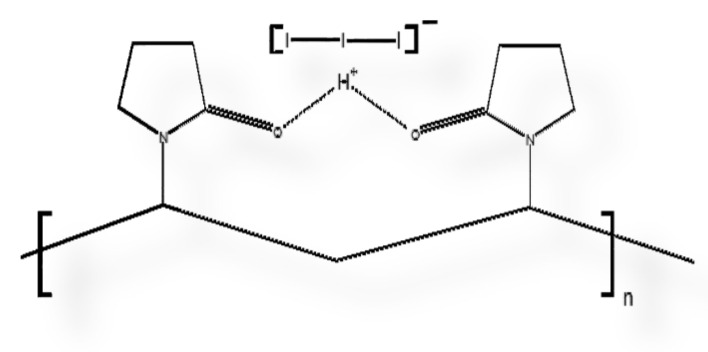
Scheme of polyvinylpyrrolidone (PVP) polymer backbone adhering triiodide ion (I_3_^−^) through hydrogen bridge, resulting in PVP-I_2_, which is actually (PVP-I_3_^−^).

**Figure 10 biomimetics-05-00045-f010:**
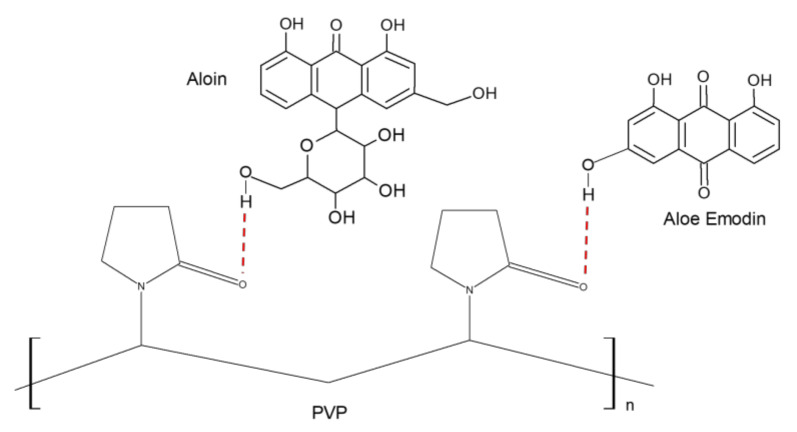
Scheme of hydrogen bonding between *Aloe vera* (AV) components with the examples Aloin and Aloe Emodin with polyvinylpyrrolidone (PVP).

**Figure 11 biomimetics-05-00045-f011:**
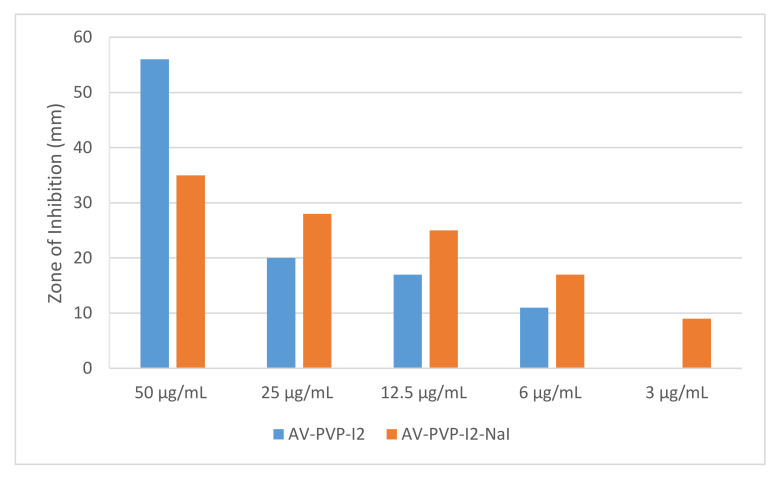
Antimicrobial disc diffusion assay of the biomaterials AV-PVP-I_2_ (blue) and AV-PVP-I_2_-NaI (orange) against *C. albicans WDCM 00054* at different dilutions.

**Table 1 biomimetics-05-00045-t001:** Antimicrobial testing of antibiotics (A), AV-PVP-I_2_ (+), AV-PVP-I_2_-NaI (++), and dilutions. ZOI (mm) against microbial strains by disc diffusion assay as dilution series.

Strain	Antibiotic	A	1^+^	2^+^	3^+^	4^+^	1^++^	2^++^	3^++^	4^++^
*S. pneumoniae ATCC 49619*	G	21	14	12	10	0	13	0	0	0
*S. aureus ATCC 25923*	G	28	25	20	19	8	20	13	11	0
*S. pyogenes ATCC 19615*	C	25	16	10	0	0	10	0	0	0
*E. faecalis ATCC 29212*	CTX	25	17	12	9	0	0	0	0	0
*B. subtilis WDCM 00003*	S	20	22	17	10	0	16	11	8	0
*P. mirabilis ATCC 29906*	G	35	0	0	0	0	0	0	0	0
*P. aeruginosa WDCM 00026*	CTX	20	13	0	0	0	9	8	0	0
*E. coli WDCM 00013*	G	23	18	13	11	0	12	10	8	0
*K. pneumoniae WDCM 00097*	CTX	35	23	14	11	0	11	8	0	0
*C. albicans WDCM 00054*	NY	16	56	20	17	11	35	28	25	17 **

Disc diffusion studies (6 mm disc impregnated with 2 mL of 50 µg/mL (1), 2 mL of 25 µg/mL (2) and 2 mL of 12.5 µg/mL (3), and 2 mL of 6 µg/mL (4) of compounds AV-PVP-I_2_ (+) and AV-PVP-I_2_-NaI (++)). G Gentamicin (30 µg/disc). CTX (Cefotaxime) (30 µg/disc). NY (Nystatin) (100 IU). C Chloramphenicol (10 µg/disc). Streptomycin (10 µg/disc) Grey shaded area represents Gram-negative bacteria. 0 = Resistant. No statistically significant differences (*p* > 0.05) between row-based values through Pearson correlation. ** Compound M4 was diluted to 3 µg/mL (5) and achieved ZOI = 9.

**Table 2 biomimetics-05-00045-t002:** Antimicrobial activity of selected antibiotics (A), AV-PVP-I_2_, AV-PVP-I_2_-NaI and their dip-coated sutures (W = 30 min after coating procedure, D = after 3 h drying). ZOI (mm) against selected microorganisms by diffusion assay.

Strain	Anti-Biotic	A	AV-PVP-I_2_ ^W^	AV-PVP-I_2_ ^D^	AV-PVP-I_2_-NaI ^W^	AV-PVP-I_2_-NaI ^D^
*S. pneumoniae ATCC 49619*	G	21	2	1	1	0
*S. aureus ATCC 25923*	G	28	7	6	6	3
*S. pyogenes ATCC 19615*	C	25	3	1	0	0
*E. faecalis ATCC 29212*	CTX	25	5	3	0	0
*B. subtilis WDCM 00003*	S	20	5	3	4	1
*P. mirabilis ATCC 29906*	G	35	0	0	0	0
*P. aeruginosa WDCM 00026*	CTX	20	0	0	0	0
*E. coli WDCM 00013*	G	23	5	2	0	0
*K. pneumoniae WDCM 00097*	CTX	35	2.5	1.5	4	2
*C. albicans WDCM 00054*	NY	16	20	9	4	2

S Diffusion studies (2.5 cm PGA sutures coated with 50 µg/mL compounds AV-PVP-I_2_ and AV-PVP-I_2_-NaI). G Gentamicin (30 µg/disc). C Chloramphenicol (10 µg/disc). S Streptomycin (10 µg/disc). CTX Cefotaxime (30 µg/disc). NY Nystatin (100 IU). Grey shaded area represents Gram-negative bacteria. 0 = Resistant. No statistically significant differences (*p* > 0.05) between row-based values through Pearson correlation.

**Table 3 biomimetics-05-00045-t003:** UV-vis absorption signals of iodine moieties in the samples AV-PVP-I_2_, AV-PVP-I_2_-NaI and AV-PVP-NaI in previous reports (nm).

Group	AV-PVP-I_2_ *	AV-PVP-I_2_-NaI *	AV-PVP-NaI *	[22,29]	[78]	[79]	[73]	[74]	[80]	[33]
I_2_	205	203			203	460	460	205 460		
I_3_^-^	290 358	290 359	290 359	290 359	288 352	297 350	290 350	290 352	290 352	
I^-^	202				193 226	293	192 225			
PVP-I_2_	305	305								310 361 395

* UV-vis absorption signals with concentration of 25 µg/mL of the methanolic extracts of AV-PVP-I_2_, AV-PVP-I_2_-NaI and AV-PVP-NaI.

**Table 4 biomimetics-05-00045-t004:** UV-vis absorption signals of sample AV in (nm).

AV Component	AV *	[Lit.]
Aloin	208, 231, 268, 296, 353	[50,68,69,82]
Aloe-Emodin	226, 257, 285	[69]
Aloesin	213, 251, 296, 360	[68]
10-O-β-d-glucopyranosyl aloenin	202, 293	[68]
Rhein	229, 295,	[83]
Pyrogallol	280,349	[84]
Hesperidin	329	[85]

* UV-vis absorption signals with concentration of 25 µg/mL of the methanolic extract AV. Grey shaded area represents phenolic compounds.

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
