# Peer review of "Facile Synthesis of Antimicrobial Aloe Vera-“Smart” Triiodide-PVP Biomaterials"

_biomimetics, 2020, doi:10.3390/biomimetics5030045_

Round 1

Reviewer 1 Report

The manuscript submitted for review concerns research on the new natural, antimicrobial materials / substances.
The scope of the research and the methods used do not raise any objections.

The text is very long - if the Authors were able to shorten the Introduction chapter a bit, it would be advisable :)
Authors tend to redundantly repeat certain information (for example, page 14, lines 4-7 and page 23 lines 24-35) and to introduce abbreviations to abbreviations (Table 1 and 2).

In the Conclusions chapter (page 26 lines 42-47), the Authors' statements are not entirely consistent with the results obtained. Please reformat them.

In the References chapter, there is no numbering - it made it very difficult to check whether all literature items in the text are cited in References and vice versa.

All comments were marked in the pdf file in the review mode. Some comments are repetitive and apply throughout the text.

Author Response

Dear reviewer,

first of all, thank you very much for your strong efforts.

I attached your valuable pdf below, while the following text is answers to your comments:

The manuscript submitted for review concerns research on the new natural, antimicrobial materials / substances.
The scope of the research and the methods used do not raise any objections. Thank you very much J

The text is very long - if the Authors were able to shorten the Introduction chapter a bit, it would be advisable :).

We were advised by another reviewer to include some information in the introduction within the 6th paragraph (page 4, line 6 and following) about novelty aspects of this work, which I did (adding line 5-11). We removed on page 4, lines 6-10 to avoid repeating explanations of comparison with our previous work. This is already discussed in the discussion on page 25, line 50 and following.

Authors tend to redundantly repeat certain information (for example, page 14, lines 4-7 and page 23 lines 24-35) and to introduce abbreviations to abbreviations (Table 1 and 2).

You are right dear reviewer, I used the original abbreviations from the text in the Table 3 and removed M3 and M4 (Page 19, line 8, Table 3 and Table 2). In the table 1 (Page 14, Line 11), I was not able to do that due to space restriction. So, instead of M3 and M4 I had to use + and ++.

In the Conclusions chapter (page 26 lines 42-47), the Authors' statements are not entirely consistent with the results obtained. Please reformat them.

Yes, it was copy-paste mistake. Page 26, lines 38-43 are the changed parts. Thank you very much again!

In the References chapter, there is no numbering - it made it very difficult to check whether all literature items in the text are cited in References and vice versa.

I am so sorry, my mistake ! I directly added the numbers. And removed the mistakes related to names, which are wrongly added. These names are from co-authors (page 10, line 12). Now changed to page 10 line 13. The references were 64 and 65.

All comments were marked in the pdf file in the review mode. Some comments are repetitive and apply throughout the text. We changed everything accordingly. Thank you very much. Will try to attach the same pdf, because we answered on the pdf as well.

Thank you very, very, very much for your immense efforts and support throughout the manuscript and your valuable comments and input. I am so grateful to follow your advice, as it will improve our manuscript !!

Thank you very much, and once again, really sorry about the mistake, not having put the numbers. I didn’t see it ! Good, that you didn’t right away refuse the manuscript, because the journal is asking for the format with the numbers. I think it happened while copy pasting the references in the final manuscript template.

Thank you also for doing more efforts by sending the pdf with comments. Really appreciated J

Thank you very much. 

Best regards

Zehra

Reviewer 2 Report

The manuscript reports on the isolation of New biomaterials consisting of modified biologically active  Aloe Vera components. Overall, the manuscripts is interesting, and well presented. It reads well with only a small number of typos. On the other hand I have a few critical comments:

  • Abstract should be modified, namely first two sentences should be removed as they suggest that the manuscript is directly related to COVID-19 pandemic subject, which is misleading to the reader.
  • The most prominent antimicrobial activities should be highlighted in the abstract
  • The aim and the novelty of the work should be clarified better in the introduction section. In particular, the paragraph 6 of the introduction section should be modified by separating the references to previous work from that of the current work. Also, the novelty aspects should be better highlighted in this paragraph.
  • Uv-Vis and IR spectroscopy – the term “peak” should be replaced with more appropriate word “band”.  In contrary to other spectroscopic methods such as NMR, MS where the signals are observed as “peaks”  the UV-Vis and IR spectra consist of the absorption “bands”. Please modify these throughout the manuscript.
  • Conclusions – the reference to the COVID-19 should be removed as the manuscript does not deal with any aspects which are directly related to this topic.

Overall the manuscript seems worth-publishing in MDPI Biomimetics after minor revisions.

Author Response

Dear reviewer,

kindly see the attachment.

Thank you so much indeed !

Best regards

Zehra

Reviewer 3 Report

The work is very interesting. The experiment was well planned and implemented. The methodology is described in detail.
The work can be accepted for printing after minor corrections.

1. The introduction is too long and should be shortened.
2. Figure 2b and 2d are not clear. You need to improve the resolution of photos.
3. Figures 9 and 10 should be corrected and drawn as required by the magazine.
4. In Figure 11 there is no description of one of the axles.
5. Conclusions should be shortened. This part should contain the most important achievements.

Author Response

Dear Reviewer,

the report with answers is attached.

Thank you so much for your valuable comments.

We are happy to improve our manuscript through your support.

Best regards

Zehra

Round 2

Reviewer 1 Report

Thank you for introducing the corrections, the only remark is "in-vitro" without the dash "in vitro" (in Abstract)
Good luck